# RD$^2$: Reward Decomposition
# with Representation Disentanglement

**Zichuan Lin**$^*$
Tsinghua University
lzcthu12@gmail.com

**Derek Yang**$^*$
UC San Diego
dyang1206@gmail.com

**Li Zhao**
Microsoft Research
lizo@microsoft.com

**Tao Qin**
Microsoft Research
taoqin@microsoft.com

**Guangwen Yang**
Tsinghua University
ygw@tsinghua.edu.cn

**Tieyan Liu**
Microsoft Research
tyliu@microsoft.com

## Abstract

Reward decomposition, which aims to decompose the full reward into multiple sub-rewards, has been proven beneficial for improving sample efficiency in reinforcement learning. Existing works on discovering reward decomposition are mostly policy dependent, which constrains diversified or disentangled behavior between different policies induced by different sub-rewards. In this work, we propose a set of novel policy-independent reward decomposition principles by constraining uniqueness and compactness of different state representations relevant to different sub-rewards. Our principles encourage sub-rewards with minimal relevant features, while maintaining the uniqueness of each sub-reward. We derive a deep learning algorithm based on our principle, and refer to our method as RD$^2$, since we learn reward decomposition and disentangled representation jointly. RD$^2$ is evaluated on a toy case, where we have the true reward structure, and chosen Atari environments where the reward structure exists but is unknown to the agent to demonstrate the effectiveness of RD$^2$ against existing reward decomposition methods.

## 1 Introduction

Since deep Q-learning was proposed by Mnih et al. [2015], reinforcement learning (RL) has achieved great success in decision making problems. While general RL algorithms have been extensively studied, here we focus on those RL tasks with multiple reward channels. In those tasks, we are aware of the existence of multiple reward channels, but only have access to the full reward. Reward decomposition has been proposed for such tasks to decompose the reward into sub-rewards, which can be used to train RL agent with improved sample efficiency.

Existing works mostly perform reward decomposition by constraining the behavior of different policies induced by different sub-rewards. Grimm and Singh [2019] propose encouraging each policy to obtain only its corresponding sub-rewards. However, their work requires that the environment be reset to arbitrary state and cannot be applied to general RL settings. Lin et al. [2019] propose encouraging the diversified behavior between such policies, but their method only obtains sub-rewards on transition data generated by their own policy, therefore it cannot decompose rewards for arbitrary state-action pairs.

In this paper, we propose a set of novel principles for reward decomposition by exploring the relation between sub-rewards and their relevant features. We demonstrate our principles based on a toy

---

$^*$Equal contribution

environment Monster-Treasure, in which the agent receives a negative reward $r_{monster}$ when it runs into the wandering monster, and receives a positive reward $r_{treasure}$ when it runs into the treasure chest. A good decomposition would be to split the reward $r$ into $r_{monster}$ and $r_{treasure}$, where only some features are relevant to each sub-reward. To be specific, only the monster and the agent are relevant to predicting $r_{monster}$. A bad decomposition could be splitting the reward into $\frac{r}{2}$ and $\frac{r}{2}$, or $r$ and $0$. The first one is not compact, in the sense that all features are relevant to both sub-rewards. The latter one is trivial, in the sense that none of the features is relevant to the $0$ sub-reward. We argue that if each of the sub-reward we use to train our agent is relevant to limited but unique features only, then the representation of sub-returns induced by sub-rewards would also be compact and easy to learn.

Motivated by the example above, we propose decomposing a reward into sub-rewards by constraining the relevant features/representations of different sub-rewards to be compact and non-trivial. We first derive our principles for reward decomposition under the factored Markov Decision Process(fMDP). Then we relax and integrate the above principles into deep learning settings, which leads to our algorithm, Reward Decomposition with Representation Disentanglement($RD^2$). Compared with existing works, $RD^2$ can decompose reward for arbitrary state-action pairs under general RL settings and does not rely on policies. It is also associated with a disentangled representation so that the reward decomposition is self-explanatory and can be easily visualized. We demonstrate our reward decomposition algorithm on the Monster-Treasure environment discussed earlier, and test our algorithm on chosen Atari Games with multiple reward channels. Empirically, $RD^2$ achieves the following:

- It discovers meaningful reward decomposition and disentangled representation.
- It achieves better performance than existing reward decomposition methods in terms of improving sample efficiency for deep RL algorithms.

## 2 Background and Related Works

### 2.1 MDP

We consider general reinforcement learning, in which the interaction of the agent and the environment, can be viewed as a Markov Decision Process (MDP)[Puterman, 1994]. Denoting the state space by $\mathcal{S}$, action space by $\mathcal{A}$, the state transition function by $P$, the action-state dependent reward function by $R$ and $\gamma$ the discount factor, we write this MDP as $(\mathcal{S}, \mathcal{A}, \mathcal{R}, P, \gamma)$. Here a reward $r$ is dependent on its state $s \in \mathcal{S}$ and action $a \in \mathcal{A}$.

$$r = \mathcal{R}(s, a) \tag{1}$$

A common approach to solving an MDP is by estimating the action-value $Q^\pi(s, a)$, which represents the expected total return for each state-action pair $(s, a)$ under a given policy $\pi$.

### 2.2 Factored MDP

Our theoretical foundation is based on factored MDP (fMDP). In a factored MDP [Boutilier et al., 1995, 1999], state $s \in \mathcal{S}$ can be described as a set of factors $s = (x_1, x_2, ..., x_N)$. In some factored MDP settings, reward function $\mathcal{R}$ can be decomposed into multiple parts where each part returns a sub-reward, or localized reward. Let $s_i$ be a fixed subset of factors in $s$, denoted by $s_i \subset s$, localized rewards $r_i$ only depend on sub-states:

$$r_i = \mathcal{R}_i(s_i, a) \tag{2}$$

and the full reward is obtained by $\mathcal{R}(s, a) = \sum_{i=1}^{K} \mathcal{R}_i(s_i, a)$.

In most environments, while the reward structure exists latently, we do not know the sub-reward functions $\mathcal{R}_i$ nor the sub-rewards $r^i$ and only the full reward $r$ is observable.

### 2.3 Reward Decomposition

Having access to sub-rewards $r_i$ can greatly accelerate training in RL [Schneider et al., 1999, Littman and Boyan, 1993, Russell and Zimdars, 2003, Bagnell and Ng, 2006, Marthi, 2007, Van Seijen et al., 2017, OpenAI et al., 2019]. Hybrid Reward Architecture (HRA) [Van Seijen et al., 2017] proposes learning multiple Q-functions, each trained with its corresponding sub-reward and showed

significant improvements compared to training a single Q-function. However, in HRA the rewards are decomposed manually. In Dota 2 [OpenAI et al., 2019], over 10 reward types associated with different parts of the state, e.g. gold, kills, mana, etc., are designed intrinsically to help the agent plan better. Reward decomposition can also be used for multi-agent settings [Russell and Zimdars, 2003].

Given the potential of utilizing sub-rewards, finding a good reward decomposition in an unknown environment becomes an important line of research. Reward decomposition seeks to find sub-rewards $r^i$ without any domain knowledge. Grimm and Singh [2019] and Lin et al. [2019] both make an assumption of policy disagreement to perform reward decomposition. Lin et al. [2019] first perform reinforcement learning jointly with reward decomposition without domain knowledge or manipulating environments. However, Lin et al. [2019] can only compute sub-rewards from sub-values for transition data generated by their own policy, making it hard to apply learned sub-rewards to downstream tasks such as training new agents.

## 2.4 Disentangled Representation

A recent line of work has argued that representations that are disentangled are an important step towards a better representation learning [Bengio et al., 2013, Peters et al., 2017, Higgins et al., 2017, Chen et al., 2016, 2018, Hsu et al., 2017]. The key idea is that a disentangled representation should separate the distinct, informative factors of variations in the data. Particularly, entropy reduction has been used for representation disentanglement in prior works [Li et al., 2019]. Different from those works, we focus on reward decomposition in RL, and learn compact representation for each sub-reward. Although we encourage the compactness and diversity of different representations for different sub-rewards, there are usually some overlap between different representations, which is different from the idea of disentangled representation. For example, in the Monster-Treasure environment, the agent information is important for representations of both $r_{monster}$ and $r_{treasure}$.

## 3 Minimal Supporting Principle for Reward Decomposition

In this section, we introduce our principles for finding minimal supporting reward decomposition under fMDP. The first principle is that the relevant features of the sub-rewards should contain as little information as possible, which implies compactness. To define relevant features formally, we first define minimal sufficient supporting sub-state. We further define K-minimal supporting reward decomposition, which directly leads to our second principle: each sub-reward should be unique in that their relevant features contain exclusive information. The second principle encourages diversified sub-rewards and features that represent different parts of the reward dynamics.

### 3.1 Minimal Sufficient Supporting Sub-state

We first consider an fMDP with known sub-reward structures. E.g., $r_{monster}$ and $r_{treasure}$ in the Monster-Treasure environment introduced in the Introduction part. Let state be composed of $N$ factors, denoted by $s = \{x_1, x_2, x_3, ..., x_N\}$ and denote the $i-$th sub-reward at state $s$ by $r_i(s)$, $i \in [1, K]$. For example, state in the Monster-Treasure environment is $\{s_{agent}, s_{monster}, s_{treasure}\}$, where $s_{agent}/s_{monster}/s_{treasure}$ represent the state of the agent/monster/treasure chest respectively. Sub-state $s_i$ is extracted from state $s$ by selecting a subset of variables. For sub-reward $r_{monster}$, the best sub-state would be $\{s_{agent}, s_{monster}\}$ because it contains only relevant information for predicting $r_{monster}$. Motivated by this observation, we define minimal sufficient supporting sub-state in definition 1.

**Definition 1.** *A sub-state $s_i \subset s$ is the minimal sufficient supporting sub-state of $r_i$ if*

$$H(s_i) = \min_{\hat{s}_i \in M_i} H(\hat{s}_i)$$

$$M_i = \{\hat{s}_i | H(r_i | \hat{s}_i, a) = \min_{\bar{s}} H(r_i | \bar{s}, a), \bar{s} \in s\}$$

*where $H(r_i | s_i, a)$ denotes conditional entropy.*

If $s_i \in M_i$ but $H(s_i) \neq \min_{\hat{s}_i \in M_i} H(\hat{s}_i)$, we refer to such sub-state as sufficient supporting sub-state.

The intuition of minimal sufficient supporting sub-state is to contain all and only the information required to compute a sub-reward. Note that $H(r_i | s_i, a)$ is not necessarily 0 because of intrinsic randomness.

## 3.2 K-Minimal Supporting Reward Decomposition

To introduce our principles for reward decomposition, we start from several undesired trivial decompositions and one specific desired decomposition in the Monster-Treasure environment discussed in the previous section.

The first trivial decomposition would be splitting the total reward into two equivalent halves, i.e. $\frac{r}{2}$ and $\frac{r}{2}$, where the minimal sufficient supporting state for both channels would be $s_1 = s_2 = s$. Another trivial decomposition is $r_1 = r$ and $r_2 = 0$ with corresponding minimal sufficient supporting sub-states $s_1 = s$ and $s_2 = \emptyset$, notice that the second channel would not contain any information. A more general case of trivial decomposition would be $r_1 = r + f(s_{agent})$ and $r_2 = -f(s_{agent})$ with corresponding minimal sufficient supporting sub-states $s_1 = s$ and $s_2 = \{s_{agent}\}$, where $f$ is an arbitrary function. The second channel does contain information but is in fact redundant. The last undesired decomposition would be $r_1 = r_{monster} + \frac{1}{2} r_{treasure}$ and $r_2 = \frac{1}{2} r_{treasure}$ where the corresponding minimal sufficient supporting sub-states are $s_1 = s$ and $s_2 = \{s_{agent}, s_{treasure}\}$. $s_{treasure}$ in $s_1$ is clearly redundant.

The ideal decomposition for the Monster-Treasure environment would be to decompose the reward $r$ into $r_{monster}$ and $r_{treasure}$, because it is a compact decomposition in which each sub-reward has a compact minimal sufficient supporting sub-state. To distinguish the ideal decomposition from the trivial ones, the first principle is that each channel should contain exclusive information that other channels do not. On top of that, the second principle is that the sum of the information contained in each channel should be minimized.

Motivated by above observation, we define K-minimal supporting sub-rewards as follows:

**Definition 2.** *Let $s_i$, $\hat{s}_i$ be the minimal sufficient supporting sub-state for $r_i$, $\hat{r}_i$ correspondingly. A set of sub-rewards $\{r_i(s)\}_{i=0}^K$ forms a K-minimal supporting reward decomposition if:*

$$\sum_i^K H(s_i) = \min_{\{\hat{r}_i\} \in C} \sum_i^K H(\hat{s}_i)$$

$$C = \left\{ \{\hat{r}_i\} | \sum_{i=1}^K \hat{r}_i = r, \hat{s}_i \subsetneq \hat{s}_j \ \forall i, j \right\}$$

Note that there could be multiple K-minimal reward decompositions, e.g. swapping two channels of a K-minimal reward decomposition will create a new one. The intuition of K-minimal supporting reward decomposition is to encourage non-trivial and compact decomposition, while no sub-state $s_i$ is a subset of other sub-state $s_j$.

## 4 RD$^2$ Algorithm

Minimal supporting principles define our ideal reward decomposition under factored MDP, where selecting factors is inherently optimizing a boolean mask over factors. However, complex environments pose more challenges in developing a practical algorithm. To be specific, the first challenge is to allow complex states such as raw images as input, rather than extracted factors. The second challenge is that estimating entropy in deep learning using either sample-based or neural estimation methods could be time-consuming. In this section we propose several techniques to overcoming these two challenges.

### 4.1 Objectives

To overcome the challenge of taking raw images as input, instead of viewing pixels as factors, we use a $H' \times W' \times N$ feature map $f(s)$ as a map of factors, each encoding regional information. Here $H'$ and $W'$ represent the height and width after convolution, and $N$ is the number of channels of feature map.

In Section 3., we assume that $s_i$ picks a fixed subset of $s$ as sub-state, which is inherently a fixed binary mask. However, in image-based RL environment, even when we are using feature map instead of raw pixels, it is not realistic to assume that the mask would be fixed for all states. This is similar to the attention mechanism, e.g. in the Monster-Treasure environment the mask would need to follow

the monster's position to extract its information. To this end, we allow the mask on the feature map to be dependent on state $s$, given by $m_i(s)$. Sub-state $\hat{s}_i$ can then be represented by

$$\hat{s}_i = f(s) \odot m_i(s) \tag{3}$$

Definition 2 implies that the final objective for a reward decomposition is reached by minimizing $\sum H(\hat{s}_i)$. Normally we would first find the minimal sufficient supporting state $s_i$ of a given reward decomposition, represented by $r_i$, then evaluate $\sum H(s_i)$. However, this objective cannot back propagate to $r_i$ since the operation of finding minimal sufficient supporting sub-state is not derivable.

To tackle this issue, we let $r_i$ be directly dependent of $\hat{s}_i$ by $r_i = g_{\theta_i}(\hat{s}_i, a)$. The first constraint for K-minimal supporting reward decomposition then leads to a straightforward objective:

$$\mathcal{L}_{sum} = (r - \sum_{i=1}^{K} g_{\theta_i}(\hat{s}_i, a))^2 \tag{4}$$

Note that $\hat{s}_i$ would always be a sufficient supporting sub-state for $r_i$, but not necessarily minimal. However, the minimal condition in definition 1 can be approximated by minimizing $H(\hat{s}_i)$, which is also the objective of K-minimal supporting reward decomposition given by definition 2. So our second objective is given by

$$\mathcal{L}_{mini} = \sum_{i=1}^{K} H(\hat{s}_i) \tag{5}$$

The above two terms are still not suffice for finding K-minimal supporting reward decomposition. The second constraint of definition 2, which is the non-trivial requirement, suggests that $s_i \subsetneq s_j, \ \forall i, j$, which is also equivalent to $H(\hat{s}_i | \hat{s}_j) > 0$ in general cases. This constraint is found critical in our experiments. Also, as an alternative, an equivalent objective according to definition 1 is $H(r_i | \hat{s}_i, a) < H(r_i | \hat{s}_j, a)$.

Instead of simply demanding inequality, we further maximize $H(\hat{s}_i | \hat{s}_j)$ or $H(r_i | \hat{s}_j, a)$ to encourage diversity between sub-states. The last objective is given by

$$\mathcal{L}_{div1} = -\sum_{i=1}^{K} \sum_{j=1, j \neq i}^{K} H(\hat{s}_i | \hat{s}_j) \tag{6}$$

or

$$\mathcal{L}_{div2} = -\sum_{i=1}^{K} \sum_{j=1, j \neq i}^{K} H(r_i | \hat{s}_j, a). \tag{7}$$

## 4.2   Surrogate Loss for Entropy Estimation

Computing $\mathcal{L}_{mini}$ and $\mathcal{L}_{div}$ requires entropy estimation. Since the state space in Atari is very large, using sampling-based entropy estimation methods is unrealistic. There exist reliable methods on neural entropy estimation, but are in general time-consuming. In our problem, we introduce approximate losses that are reasonable and convenient in our setting.

**Approximating $H(\hat{s}_i)$**   Recall that $H(cX) = H(X) + log(|c|)$ and $H(X|cY) = H(X|Y)$ when $c$ is a constant and $c \neq 0$. Since we let $m_i \in (0, 1)^N$ and that $\hat{s}_i = f(s) \odot m_i(s)$, an empirical estimation for $H(\hat{s}_i)$ can be derived:

$$H(\hat{s}_i) \approx H(f(s)) + \sum_{l=1}^{N} log(m_{i,l}(s)) \leq H(f(s)) + log(\sum_{l=1}^{N} m_{i,l}(s)) \tag{8}$$

where $N$ is the size of the feature map. Note that if $m$ is fixed, the first approximation becomes equality. The last inequality gives an upper bound that resolves numerical issues of taking $log$ of a small float. Since the entropy of the feature map $H(f(s))$ is irrelevant to the mask, we can optimize $H(s_i)$ approximately by minimizing the second term:

$$\mathcal{L}_{mini} = \sum_{i=1}^{K} log(\sum_{l=1}^{N} m_{i,l}(s)) \tag{9}$$

**Approximating** $H(\hat{s}_i|\hat{s}_j)$  Inspired by the method for estimating $H(\hat{s}_i)$, we propose using an intuitive approximate loss for $H(\hat{s}_i|\hat{s}_j)$ that resembles $\mathcal{L}_{mini}$:

$$\mathcal{L}_{div1} = -\sum_{i=1}^{K} \sum_{j=1,j\neq i}^{K} log(\sum_{l=1}^{N} ReLU(m_{i,l}(s) - m_{j,l}(s))). \qquad (10)$$

To further explain the intuition behind $\mathcal{L}_{div1}$, consider a factored MDP where a factor is either chosen or not chosen for each sub-state. Note that a factor $x_k$ will contribute to $H(\hat{s}_i|\hat{s}_j)$ only if $x_k$ is chosen by $\hat{s}_i$ but not chosen by $\hat{s}_j$, i.e. $m_{i,k} = 1$ **and** $m_{j,k} = 0$. A simple way to extend this logical expression to real values is to use $ReLU(m_{i,k} - m_{j,k})$.

**Approximating** $H(r_i|\hat{s}_j, a)$  Estimating $H(r_i|\hat{s}_j, a)$ could be complicated in general, however if we assume that $H(r_i|\hat{s}_j, a)$ is only related to the logarithm of its variance (e.g. Gaussian distribution), i.e. $H(r_i|\hat{s}_j, a) \sim log(Var(r_i|\hat{s}_j, a))$, then a surrogate objective can be derived.

Note the definition of variance $Var(r_i|\hat{s}_j, a) = \mathbb{E}\left[r_i - \mathbb{E}(r_i|\hat{s}_j, a)\right]^2$. To obtain an estimation for $\mathbb{E}(r_i|\hat{s}_j, a)$, we use a network $\hat{r}_i = g_{\theta_{ij}}(\hat{s}_j, a)$ and minimize $MSE(r_i, \hat{r}_i)$ over parameter $\theta_{ij}$. We can then use $\hat{r}_i$ as an estimation for $\mathbb{E}(r_i|\hat{s}_j, a)$ and $MSE(r_i, \hat{r}_i)$ as an approximation for $Var(r_i|\hat{s}_j, a)$. Thus maximizing $MSE(r_i, \hat{r}_i)$ over $\hat{s}_j$ will be equivalent to increasing $log(Var(r_i|\hat{s}_j, a))$, i.e. $H(r_i|\hat{s}_j, a)$.

$$\mathcal{L}_{div2} = -\sum_{i=1}^{K} \sum_{j=1,j\neq i}^{K} log(\min_{\theta_{ij}}(g_{\theta_i}(\hat{s}_i, a) - g_{\theta_{ij}}(\hat{s}_j, a))^2). \qquad (11)$$

$\mathcal{L}_{div2}$ penalizes information in $\hat{s}_j$ that is related to $r_i$, which would enforce different channels to contain diversified information.

The final objective of RD$^2$ is given by:

$$\mathcal{L} = \alpha\mathcal{L}_{sum} + \beta\mathcal{L}_{mini} + \gamma\mathcal{L}_{div} \qquad (12)$$

where $\mathcal{L}_{div}$ has two alternatives and $\alpha/\beta/\gamma$ are coefficients. We provide the pseudo code of our algorithm in Appendix 1.

# 5  Experiment

In our experiments, we aim to answer the following questions: (1) Can RD$^2$ learn reward decomposition? (2) Does RD$^2$ learn meaningful mask on state input? (3) How does RD$^2$ perform in terms of using decomposed rewards to improve sample efficiency?

## 5.1  Toycase

In this section, we test RD$^2$ with mini-gridworld [Chevalier-Boisvert et al., 2018], configured to the Monster-Treasure environment discussed earlier as shown in Figure 1. In this environment, $r_{treasure} = 2$ when the agent (red triangle) finds the treasure (green grid), otherwise $r_{treasure} = 0$. The agent also receives a reward of $r_{monster} = -2$ when it collides with the moving monster (blue ball), otherwise $r_{monster} = 0$. Note that if the agent finds the treasure and collides with the monster at the same time, the reward $r = r_{treasure} + r_{monster}$ will also be 0.

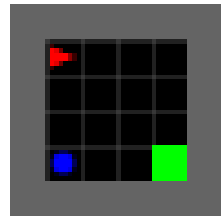

Figure 1: Monster-Treasure

The coordinates of the objects are extracted into factors and are given by $\{agent_x, agent_y, monster_x, monster_y, treasure_x, treasure_y\}$. The network takes as input the factors and the action, and is trained with equation 12 using the $\mathcal{L}_{div1}$ variant. The mask in this case is trainable but does not depend on the input. Note that only $r = r_{treasure} + r_{monster}$ is used as a training signal.

We find that RD$^2$ is able to completely separate $r_{treasure}$ and $r_{monster}$ trained only with $r$. As shown in Figure 2, the MSE loss for $r_{treasure}$ and $r_{monster}$ eventually converges to 0. The mask gradually converges to the optimal mask, where $\hat{s}_1 = \{agent_x, agent_y, treasure_x, treasure_y\}$ and $\hat{s}_2 = \{agent_x, agent_y, monster_x, monster_y\}$.

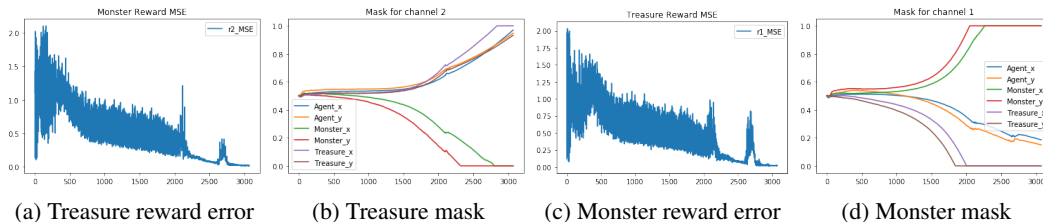

| (a) Treasure reward error | (b) Treasure mask | (c) Monster reward error | (d) Monster mask |

Figure 2: Monster-Treasure training curves

| $r_{treasure}$ | $r_{monster}$ | $r$ | predicted $r_{treasure}$ | predicted $r_{monster}$ | predicted $r$ |
|---|---|---|---|---|---|
| 2.00 | 0.00 | 2.00 | 1.85 | 0.13 | 1.99 |
| 0.00 | -2.00 | -2.00 | -0.14 | -1.86 | -2.00 |
| 0.00 | 0.00 | 0.00 | 0.11 | -0.15 | -0.04 |
| 2.00 | -2.00 | 0.00 | 1.92 | -1.84 | 0.08 |

Table 1: Example of reward decomposition on Monster-Treasure

In Monster-Treasure, there are two possible $(s, a)$ pair that would receive a reward of 0. One is $r_{treasure} = 0$ and $r_{monster} = 0$, which is trivial. The second one is $r_{treasure} = 2$ and $r_{monster} = -2$, meaning that the agent finds the treasure but bumps into the monster at the same time. It is notable that while $r$ does not show the difference between those two cases, RD$^2$ is capable of telling the difference even when the total rewards are both 0, since both $r_{treasure}$ and $r_{monster}$ are predicted accurately as shown in Table 1.

One specific observation due to continuous masking between 0 and 1 is that, although both channel masks have non-zero values on agent related factors, values of channel 2 are significantly larger than values of channel 1 due to $\mathcal{L}_{div1}$. However, as long as the value does not go to zero, we can consider that channel 1 views agent coordinates as required factors.

## 5.2 Atari Domain

We also run our algorithm on a more complicated benchmark called Atari. Following Lin et al. [2019], We experiment with the Atari games that have a structure of multiple reward sources. We first present the results of reward decomposition and visualize the trained masks using saliency maps on several Atari games, and then show that our decomposed rewards can accelerate the training process of existing RL algorithms. We show that RD$^2$ achieves much better sample efficiency than the recently proposed reward decomposition method DRDRL [Lin et al., 2019] and Rainbow [Hessel et al., 2018].

**Reward decomposition.** We demonstrate that RD$^2$ can learn meaningful reward decomposition on Atari games which has multiple-reward structure. Figure 3 shows the results. In the game *UpNDown*, the agent receives a reward of 3 when it hits a flag, and receives a reward of 2 when it jumps on another car. We show that our algorithm can decompose these two reward signals into two channels — when it jumps on another car, the first channel is activated and outputs a reward of 2; when it hits a flag, the second channel will dominate the reward prediction and output a reward close to 3.

**Visualization.** To better understand how our algorithm works, we visualize the saliency map [Simonyan et al., 2013] by computing the absolute value of the Jacobian $\frac{\partial r_i}{\partial s}$ for each channel ($i = 1, 2$) in Figure 4 for the games *UpNDown* and *Gopher*. We find that RD$^2$ successfully learns meaningful state decomposition. In *UpNDown* (top row), the first channel (blue) attends to the flag when the agent hits it (top left), while the second channel (pink) attends to other cars which the agent jumps on (top right).

In *Gopher* (bottom row), the agent receives a reward of 0.15 when it fills the hole in ground(bottom left) and a reward of 0.8 when it catches a gopher (bottom right). We notice that RD$^2$ learns a saliency map that accurately distinguishes these two cases. The first channel (blue) attends to the ground and predicts the 0.15 reward while the second channel (pink) attends to the gopher and predicts the 0.8

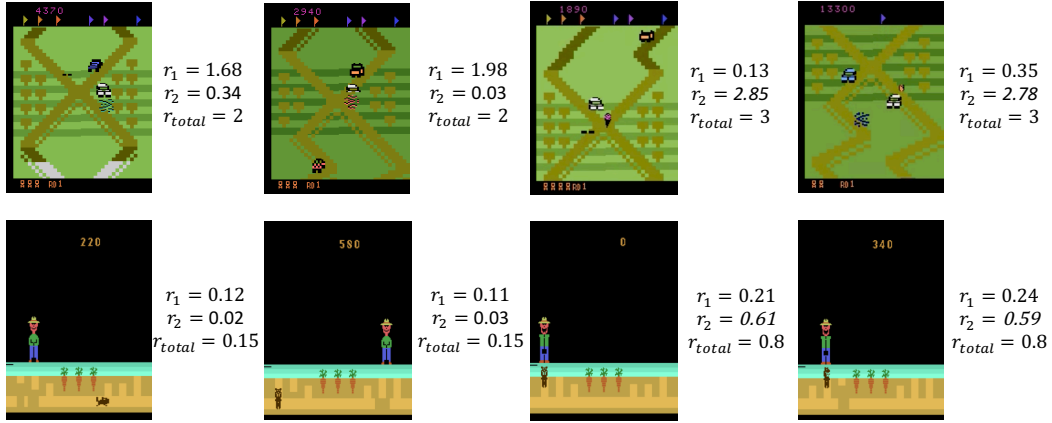

Figure 3: Reward decomposition results.

reward. We also find that with the help of dynamic mask, the second channel (pink) always have attention on the gopher.

**Joint training performance.** We now simultaneously train the sub-reward function and the sub-Q network and use the decomposed reward to directly train the sub-Q networks for each channel as in Lin et al. [2019], Van Seijen et al. [2017]. In brief, we train multiple Q networks and introduce an additional sub-Q TD error defined by

$$\mathcal{L}_{TD_i} = [Q_i(s,a) - r_i - \gamma Q_i(s',a')]^2 \qquad (13)$$

Note that we use global action $a' = \operatorname{argmax}_a \sum_i Q_i(s,a)$ instead of local actions $a'_i = \operatorname{argmax}_a Q_i(s_{t+1}, a)$ to assure unchanged optimal Q-function. For a detailed version of combining $RD^2$ with Q-learning, please refer to Appendix A.

Q-learning combined with $RD^2$ shows great improvements in sample efficiency compared with both Rainbow and DRDRL as shown in Figure 5. At early epochs the curves of $RD^2$ are below baselines due to noise in sub-reward signals. But once the reward decomposition part was partly trained, it accelerates an agent's learning process significantly.

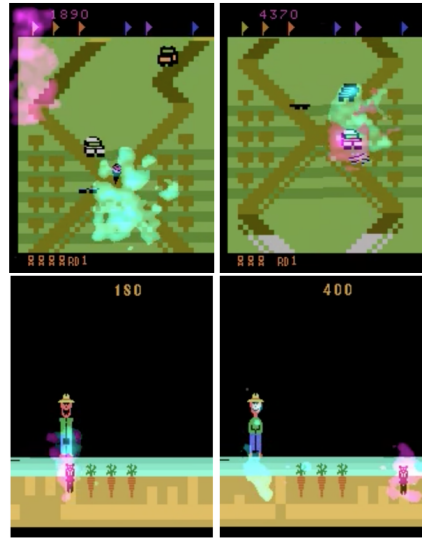

Figure 4: Saliency map visualization.

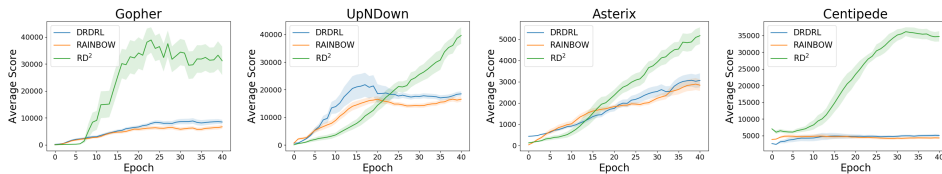

Figure 5: Joint training performance on Atari games. Each curve is averaged by three random seeds.

## 6 Discussion and Conclusion

In this paper, we propose a set of novel reward decomposition principles which encourage sub-rewards to have compact and non-trivial representations, termed $RD^2$. Compared with existing methods, $RD^2$

is capable of decomposing rewards for arbitrary state-action pairs under general RL settings and does not rely on policies. Experiments demonstrate that $RD^2$ greatly improves sample efficiency against existing reward decomposition methods. One possible explanation for the performance of $RD^2$ is its relation to learning compact state representation. Each learned sub-reward is dependent only on a subset of the state, allowing the corresponding sub-value to also depend on a subset of the state and thus learn a compact representation for such sub-values. Therefore, $RD^2$ naturally has a closer connection to learning compact representation for sub-values and speed up RL algorithms.

In the future, we will explore reward decomposition under multi-agent RL setting. The state in multi-agent RL may have natural graph structure modeling agents' interaction. We will explore how to leverage such structure for a better reward decomposition.

## Broader Impact

Reinforcement learning has a wide range of applications in real life. In board games [Schrittwieser et al., 2019], RL has shown that it has the potential to beat human and therefore provide valuable insights. In optimal control, RL has also been widely used as a search policy that guarantees convergence. In general planning problems such as traffic control or recommendation system, introducing RL is also an active line of research.

Reward decomposition has a lot of potential impacts, especially in multi-agent setting, where each agent should obtain a portion of the total reward, and in interpretation-required problems such as recommendation system. $RD^2$ is capable of both decomposing rewards into sub-rewards, and on top of that provide meaningful interpretation due to disentangled representation. Integrating $RD^2$ with those settings would provide benefits to both training aspects and interpretability aspects.

However, the rise of autonomous analytic algorithms will inevitably decrease the demand for human data analysts.

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
