[Supplementary Material]

## A  Algorithm

Algorithm 1 provides pseudo code for RD$^2$ on the Atari environment, which learns sub-Q network with jointly learned reward decomposition. Note that RD$^2$ can plug in any Q-learning based methods. We found that the second variant of $\mathcal{L}_{div}$ works better in Atari. At each time step, we first interact with environments, collect samples in replay buffer (Line 3 to 6). We then train the sub-reward prediction network to predict the total reward with minimal sufficient supporting sub-state (Line 9). We also train the auxiliary prediction network to predict sub-reward $r_i$ using sub-state $\hat{s}_j$ (Line 10) to compute $\mathcal{L}_{div2}$. After that, we update the mask network $m_i$ to encourage diversity between sub-states (Line 13).

To train our RL agent, we first perform standard Q-learning using TD error (Line 16) with the full reward. Simultaneously, we use the decomposed sub-rewards to directly train sub-Q network with a global action (Line 20, 21).

---

**Algorithm 1** RD$^2$: Reward Decomposition with Representation Decomposition

---

1: Initialize replay buffer $\mathcal{D}$, the parameters of sub-Q network $\phi_i$, sub-reward prediction network $\theta_i(i = 1, 2, ..., K)$, auxiliary prediction network $\theta_{ij}(i \neq j)$, and mask network $m_i(i = 1, 2, ..., K)$.
2: **for** time step t **do**
3:     Receive observation $s_t$ from environment.
4:     Select action using $\epsilon$-greedy policy $a_t \leftarrow \mathrm{argmax}_a \sum_i Q_{\phi_i}(s_t, a)$.
5:     Take action $a_t$, receive reward $r_t$ and next state $s_{t+1}$
6:     Append $(s_t, a_t, r_t, s_{t+1})$ to $\mathcal{D}$.
7:     **if** $t$ mod $n_{mini} == 0$ **then**
8:         Sample training experiences $(s, a, r, s')$ from $\mathcal{D}$.
9:         Update parameters $\theta_i$ to minimize the $\mathcal{L}_{sum}$ in Eq. 4 and $\mathcal{L}_{mini}$ in Eq. 9.
10:        Update parameters $\theta_{ij}$ in Eq. 11: $\min_{\theta_{ij}}(g_{\theta_i}(\hat{s}_i, a) - g_{\theta_{ij}}(\hat{s}_j, a))^2$
11:     **if** $t$ mod $n_{div} == 0$ **then**
12:         Sample training experiences $(s, a, r, s')$ from $\mathcal{D}$.
13:        Update parameters $m_i$ to minimize $\mathcal{L}_{div2}$ in Eq. 11.
14:     **if** $t$ mod $n_{update} == 0$ **then**
15:         Sample training experiences $(s, a, r, s')$ from $\mathcal{D}$.
16:        Perform standard Q-learning to update agent's parameters $\phi$ to minimize TD error
17: $$\phi_i \leftarrow \phi_i - \eta_1 \nabla_{\phi_i} \left( \sum_i Q_{\bar{\phi}_i}(s, a) - (r + \gamma \max_{a'} \sum_i Q_{\phi_i}(s', a')) \right)^2, \forall i$$
18:     **if** $t$ mod $n_{subq} == 0$ **then**
19:         Sample training experiences $(s, a, r, s')$ from $\mathcal{D}$.
20:        Compute next action $a' = \mathrm{argmax}_{a'} \sum_i Q_{\phi_i}(s', a')$
21:        Update parameters of sub-Q network $\phi_i$ with decomposed reward $r_i = g_{\theta_i}(\hat{s}_i, a)$
22: $$\phi_i \leftarrow \phi_i - \eta_2 \nabla_{\phi_i} \left( Q_{\bar{\phi}_i}(s, a) - (r_i + \gamma Q_{\phi_i}(s', a')) \right)^2, \forall i$$

---

## B  Hyper-parameters

We build our code using the supplied implementation of [Castro et al., 2018]. For all experiments we use $K = 2$. However, $K$ could vary depending on the games we choose. Following Castro et al. [2018], we use $\eta_1 = 6.25e - 5$. We use a large learning rate ($\alpha = 10 \times \eta_1$) to minimize $\mathcal{L}_{sum}$. We sweep the learning rate $\beta, \gamma, \eta_2$ in $\{1.0, 0.1, 0.01, 0.001, 0.0001, 0.00001\} \times \eta_1$ and finally choose $\beta = 0.0001 \times \eta_1$, $\gamma = 0.1 \times \eta_1$, $\eta_2 = 0.0001 \times \eta_1$. In RD$^2$, we update parameters with $n_{mini} = 4, n_{div} = 16, n_{update} = 4, n_{subq} = 4$. We use Adam [Kingma and Ba, 2014] to optimize all parameters.

## C  Ablation Study

To investigate the contribution of each loss term in algorithm 1, we we compare three variants of RD$^2$: (1) RD$^2$ without $\mathcal{L}_{sum}$; (2) RD$^2$ without $\mathcal{L}_{mini}$; (3) RD$^2$ without $\mathcal{L}_{div2}$. As shown in Figure 6, when

we drop the $\mathcal{L}_{sum}$ term, RD$^2$ is equivalent to learn with randomly decomposed reward. Therefore, the performance deteriorates dramatically. When we drop the diversity encouraging term $\mathcal{L}_{div2}$, we get the trivial reward decomposition, which is not helpful to accelerate the training process. Finally, we find that the minimal sufficient regularization term $\mathcal{L}_{mini}$ mainly contributes to the later training process.

Figure 6: Ablation study

## D    Network Architecture

Figure 7 shows the diagram of RD$^2$ to demonstrate the workflow. $r_i$ can then be plugged into any Q-learning algorithm with multiple sub-Q functions. Note that only one of $\mathcal{L}_{div1}$ or $\mathcal{L}_{div2}$ is required. In our toy experiment, we use $\mathcal{L}_{div1}$. In Atari, we use $\mathcal{L}_{div2}$.

Figure 7: RD$^2$ work flow.

Figure 8 shows the detailed network architecture. Multiple arrows indicate different network for each of the $K$ reward channels.

Figure 8: Network architecture of RD$^2$.