[Reviews · NeurIPS 2020]

Review 1

Summary and Contributions: This paper proposes general principles for useful reward decompositions. This corresponds generically to an algorithm they propose, which finds sub-components of state with low entropy that are predictive of reward. Empirically, this reward decomposition leads to high performance on a select number of Atari games.

Strengths: > The proposed ideas for reward decomposition heuristically seems to be reasonable and sound > That the agent is able to learn interpretable decompositions on large-scale domains like Atari games like UpnDown and Gopher is very encouraging, and that these decompositions lead to massive performance gains deserves merit .

Weaknesses: > The proposed framework seems to be fundamentally limited by the fact that it is “dynamics-blind”, in that it does not factor in any environment dynamics into context for reward or state decompositions. > I would like to see more analysis about the results on Atari (Figure 5), as the margin by which the proposed method outperforms Rainbow makes me a little skeptical as to what’s exactly happening here. The specific choice of Atari games here (Gopher, UpNDown, Asterix, Centipede) is non-standard to the best of my knowledge, and it’s a well-known issue that hyperparameter tuning on a small set of games in Atari can demonstrate non-representative results on the remaining Atari games (e.g. Ali Taiga et al, 2019).

Correctness: Yes, to the best of my knowledge.

Clarity: The paper is organized well, and the use of the recurring Monster-Treasure environment throughout the text grounds the paper nicely.

Relation to Prior Work: The paper positions itself well, as best my knowledge of the related work. I’m not very familiar with the closest related works in this area.

Reproducibility: Yes

Additional Feedback: Update: I have read the rebuttal, which has addressed my concerns about the choice of entropy as measure for sufficiency, the choice of Atari games, and downstream use of the Q function. I hope that these points are addressed in the final manuscript. I have updated my score to a 7. > Why is entropy the chosen way to measure sufficiency, as compared to other measures, e.g. variance? As also you mention in the text, measuring and minimizing entropy is relatively difficult, as compared to variance or other measures of randomness. Using some quantity like variance also would obviate the need to introduce all the additional approximations in the text. > How exactly is this reward decomposition used downstream? I do see details in the appendix, but a description in the text would make the text more clear. It seems to me that while reward decompositions cannot be incorporated cleanly into Q-learning > How were the games for Atari chosen? Minor details: > L186, Atari is mentioned out of the blue w/out any context > For a factored MDP, is the transition probability also decomposed? (Sec 2.2)


Review 2

Summary and Contributions: This paper proposes a method for decomposing rewards into parts that better expose the structure of the task, thereby allowing for more efficient credit assignment and learning. In particular, the method learns masks of the state representation, from which reward parts are calculated. The proposed objective attempts to reconstruct the true reward accurately, while minimising the size of the masks and making them independent by reducing their overlap or increasing the discrepancy between the predicted sub-rewards. In an illustrative toy task as well as several Atari games, the authors’ method decomposes the rewards sensibly and learns efficiently.

Strengths: Reward decomposition is an interesting problem, and I like this overall approach. The guiding principles are intuitive: rewards should be reconstructed accurately using as little information as possible, and those information sets should be distinct so as to capture semantically different causes of reward. Although the approximations of entropy measures feel a bit stilted, the translation of the guiding principles into practical objectives leads to quite reasonable and simple loss functions. The results on the toy task are clearly illustrative of the method, and the results on Atari look good, as do the nice saliency visualisations.

Weaknesses: While the high level ideas are explained quite clearly in words, I find the mathematical exposition of the definitions 1 and 2, and the subsequent derivations from them, a bit opaque. I would encourage the authors to further clarify the interpretation of each new piece of the definition (e.g. describe the set M_i and the set C in words). Your readers will thank you! I think the explanations for the approximate objectives are a little weak. In particular, L_div1 is motivated purely by analogy to L_mini, and L_div2 is difficult for me to understand. What is theta_ij and how is it minimised in the loss? Why does the min approximate the variance? Empirically, it would be nice to see the effect of each term in the loss, and how the result can or does degenerate with any piece missing. I would encourage the authors to report frames observed by the algorithms rather than ‘Epochs’. I'd also appreciate perhaps a bit more discussion of the question of when and how to move beyond K=2. Would the algorithm work fine (e.g. create empty masks predicting zero reward) if there were more slots than reward sources?

Correctness: I believe the claims are correct, bar perhaps the statement of equation (11) which I struggled to parse.

Clarity: The paper is reasonably clear overall (see above)

Relation to Prior Work: There is some useful discussion of related work, which I am not extremely familiar with. I think it might be valuable to also briefly mention the relation to temporal return decomposition (e.g. RUDDER from Arjona-Medina et al.), in contrast to within-timestep reward decomposition, since these seem complementary.

Reproducibility: Yes

Additional Feedback: Post-rebuttal edit =============== The rebuttal makes a good move towards clarifying the definitions and approximations used. I also appreciate the ablations for the different loss components and further implementation details. As a result, I have increased my score. However, I would encourage the authors to very carefully spell out each detail (e.g. entropies over exactly what distributions) and walk through the approximations in even more detail in the final version of this work. I believe this is good work that deserves a clear exposition.


Review 3

Summary and Contributions: This paper proposes an automatic reward decomposition method which associates each sub-reward with a subset of related state features. The state set decomposition is achieved by two principles -- minimal supporting information and exclusive/unique information. Experiments demonstrate that the proposed method is more sample efficient than other reward decomposition methods. The main contribution is converting a reward decomposition problem to a representation decomposition problem and employing two compact and unique information constraints to obtain the decomposition.

Strengths: + The main advantage of the proposed method over other reward decomposition ones is that the decomposition is automatic. + It can be applied to general RL settings no relying on policies. + The idea of converting a reward decomposition problem to a representation decomposition problem is interesting. + The proposed method is more sample efficient than other reward decomposition methods and the experiments support this claim.

Weaknesses: - The proposed algorithm involves three entropy estimation. However, entropy estimation is time consuming, even though the authors simplified the estimation by approximating loss. - It is not clear how sensitive the proposed method is to hyperparameters or initial conditions.

Correctness: OnP2L66, the rewards should depend on both state and action, not just "sub-state".

Clarity: The paper is well written and easy to follow.

Relation to Prior Work: The paper discussed how this work differs from previous contributions. There is a missing reference. The idea of employing entropy reduction for representation decomposition (splitting representation into semantic and syntactic ones) has been proposed and applied to compositional language learning by Li et al 2019. Li, Y., Zhao, L., Wang, J., and Hestness, J., "Compositional generalization for primitive substitutions", EMNLP 2019.

Reproducibility: Yes

Additional Feedback: Some ablation study to evaluate the contribution of each loss and convergence analysis would strengthen the paper. The paper is built upon HRA which optimizes each sub-reward independently. It is better to improve the loss function to achieve optimal reward/policy. [After rebuttal]: I've read the author rebuttal and the other reviewers' comments. The authors have addressed most of my concerns, and I would be in favor of acceptance.


Review 4

Summary and Contributions: This paper proposes to facilitate reinforcement learning agent training through decompose rewards and features. In this work, authors assume rewards are latently composed of multiple sub-rewards and features are accordingly divided into different subsets so that one feature subset decides one sub-reward. Authors looks for optimal division of features so that each subset can predict one sub-reward accurately while each subset is compact and redundancy between subsets are as small as possible. They use various entropy to measure the quality of subset division with respect to sub-reward prediction and subset compactness. As the entropy computation is costly, authors propose a few approximation.

Strengths: Proposed method relies on the assumption that the reward is the sum of a few sub-rewards and features can be decomposed according to different sub-rewards. I think this is a reasonable assumption for many complex real world application. The idea of finding the subset for each sub-rewards and keeping each subset independent also makes sense. Empirical experiments basically show proposed method manages to isolate feature subsets for each sub-rewards and the policy performance is better than compared baselines.

Weaknesses: For the definition of supporting sub-state, I understand the goal is to minimal features to predict the sub-reward and I also agree using conditional entropy to measure the quality of prediction with the subset, but the usage of entropy of features is not very clear to me. I would recommend more explanation here. The same concern also exists in definition 2. My another concern come from section 4.2. I would recommend some derivation on these assumption, I think it is better at least to include them in supplementary because they are not very straightforward. 1. In H(s[i]), the property of H(cX)=H(X)+log(|c|) is used, but this holds because c is a constant scalar, but s[i] = f(s) * mi[s] is element-wise multiplication. Is this approximation still reasonable? 2. In H(s[i]|s[j]), the ReLU comes out without sufficient explanation. ReLU does not look to appear in the original definition of the entropy. 3. In H(r|s,a), I agree it can be a good assumption to focus on the variance. But the approximation seems to only look into difference of prediction, I would recommend more explanation on this.

Correctness: My most concern about correctness lies on some definitions and approximations, they are listed above.

Clarity: Overall, the writing of the paper is clear. But I would recommend some more details: the proposed method is based on Q-learning method, some description about this part, especially the training pipeline and how various sub Q-networks are combined and applied, should be included in the main text (I would recommend at least a concise one if space is limited).

Relation to Prior Work: Authors list various previous methods and described them clear in related work part. But I have a minor suggestion: as attention mechanism is mentioned and authors state that proposed method is similar to attention mechanism, it would be helpful to see a comparison to attention mechanism.

Reproducibility: Yes

Additional Feedback: Thanks for author feedback. My concern about using entropy of features is responded and it looks good to me now. But in author feedback, I cannot see mathematical derivation of H(ŝ[i]|ŝ[j]) and H(r[i]|ŝ[j], a), so I still have concern about these derivation. Therefore I would adjust my score to 6.

[Author Response · NeurIPS 2020]

1. We thank all reviewers for valuable comments. We commit to improving clarity of definitions/approximations/algorithm
2. details and add more discussions on related works in the camera-ready version.

3. **Usage of entropy:** Entropy is used to measure *sufficiency*, *compactness* and *uniqueness*. *Sufficiency* is measured by
4. $H(r_i|s_i,a)$ in def.1, where the sufficient sub-state set $M_i$ represents all sub-states $\hat{s}_i$ that are as informative as the
5. whole state $s$ in terms of inferring $r_i$. *Compactness* is measured by $H(s_i)$ in def.1&2, where $C$ represents all sets
6. of sub-rewards(and corresponding sub-states) that is non-trivial. *Uniqueness*(/diversity) is measured by $H(s_i|s_j)$,
7. and $H(r_i|s_j,a)$ as an alternative. One may argue that, it is easier to use feature number to capture *compactness* and
8. *uniqueness*, for example using $|s_i - s_i \cap s_j|$ to capture diversity(/uniqueness). This is a good and simple formulation
9. under factored MDP in Section 3 when all features are independent. However, for features learnt by networks,
10. independence is not guaranteed and even when $m_i$ and $m_j$ does not overlap, the mutual information between $s_i$ and
11. $s_j$ could still be high. The usage of entropy ($H(s_i|s_j)$ and $H(r_i|s_j,a)$) allows us to discourage such case while
12. $|s_i - s_i \cap s_j|$ cannot.

13. **Explanation of $L_{div1}$:** $L_{div1}$ computes the sum of $H(\hat{s}_i|\hat{s}_j)$, which can be interpreted as randomness of sub-state $\hat{s}_i$
14. given sub-state $\hat{s}_j$. To further explain the intuition behind, consider a factored MDP where a factor is either chosen or not
15. chosen for each sub-states. Note that a factor $x_k$ will **only** contribute to $H(\hat{s}_i|\hat{s}_j)$ if $x_k$ is chosen by $\hat{s}_i$ and not chosen
16. by $\hat{s}_j$, i.e. $m_{i,k}=1$ **and** $m_{j,k}=0$. A simple way to extend this boolean expression is to use $ReLU(m_{i,k}-m_{j,k})$. We
17. admit that the approximation $L_{div_1}$ for $H(s_i|s_j)$ does not deal with the correlated case of $s_i$ and $s_j$ as well as $L_{div2}$,
18. which may explain the good performance of $L_{div2}$ over $L_{div_1}$ in Atari Games where the feature could be correlated
19. rather than independent as in well-defined factored MDP (e.g. our toy case).

20. **Explanation of $L_{div2}$:** The usage of variance to approximate entropy was discussed in L203. Note the definition
21. of variance $Var(r_i|\hat{s}_j,a) = \mathbb{E}\left[r_i - \mathbb{E}(r_i|\hat{s}_j,a)\right]^2$. To obtain an estimation for $\mathbb{E}(r_i|\hat{s}_j,a)$, we use a network $\hat{r}_i =$
22. $g_{\theta_{ij}}(\hat{s}_j,a)$ and minimize $MSE(r_i,\hat{r}_i)$ over parameter $\theta_{ij}$. Then we can use $\hat{r}_i$ as an estimation for $\mathbb{E}(r_i|\hat{s}_j,a)$ and
23. $MSE(r_i,\hat{r}_i)$ as a surrogate for $Var(r_i|\hat{s}_j,a)$ and maximize $MSE(r_i,\hat{r}_i)$ over $\hat{s}_j$ to increase variance/entropy. We
24. apologize for the ambiguity and will refine it in the camera-ready version.

25. **Downstream sub-Q learning:** The detailed version of RD$^2$ algorithm can be found in Appendix A. In brief, sub-Q
26. functions are trained with both full reward TD **and** sub-reward TD. The usage of global action $a_{t+1}$ instead of local
27. actions (i.e. $a_{t+1,i} = argmax_a Q_i(s_{t+1},a)$) assures invariant optimal Q-function $Q^*$.

28. **Ablation study for each loss term:** To investigate the contribution of each loss term, we
29. show that ablative performance. Specifically, we compare three variants of RD$^2$: (1) RD$^2$

UpNDown

30. without $\mathcal{L}_{sum}$ in Eq.4; (2) RD$^2$ without $\mathcal{L}_{mini}$ in Eq.5; (3) RD$^2$ without $\mathcal{L}_{div2}$ in Eq.7. As
31. shown in Figure 1, when we drop the $\mathcal{L}_{sum}$ term, RD$^2$ is equivalent to learn with randomly
32. decomposed reward. Therefore, the performance deteriorates dramatically. When we drop
33. the diversity encouraging term $\mathcal{L}_{div2}$, we get the half-half reward decomposition, which
34. is not helpful to accelerate the training process. Finally, we find that the minimal sufficient
35. regularization term $\mathcal{L}_{mini}$ mainly contributes to the later training process.

Figure 1: Ablation study.

36. ***To Reviewer 1:*** **Q1**: *Dynamics blind.* **A1**: Decomposing dynamics is also an interesting topic that we would love to
37. look into, however it may require stricter assumptions on the environment. **Q2**: *How were the games for Atari chosen?*
38. **A2**: We follow prior work [Lin et al.'19] and test our algorithm on the Atari games that have multiple sources of reward.
39. We will run our algorithm in more environments and provide the results in Appendix.

40. ***To Reviewer 2:*** **Q1**: *Beyond K=2.* **A1**: We found that in environments with more than two reward sources, using K>2
41. will achieve better performance. Moving beyond prior info about K, self-tuning K would be an interesting future work.

42. ***To Reviewer 3:*** **Q1**: *About the runtime of estimation of approximating loss.* **A1**: Despite the estimation of approximat-
43. ing loss, our efficient implementation can train at roughly 80% of Rainbow's speed. **Q2**: *Sensitivity to hyperparameters.*
44. **A2**: We provide the hyperparameter search range in appendix B. In practice, we found that our algorithm can work well
45. if the value of hyperparameters are in a reasonable range. For example, on one hand, since the sub-Q loss and $\mathcal{L}_{mini}$
46. serve as regularization terms, we set their corresponding learning rate to a relatively small value; on the other hand, we
47. keep the learning rate of $\mathcal{L}_{sum}$ and $\mathcal{L}_{div2}$ in the same scale of original Rainbow. Overall, our algorithm is not sensitive
48. to the hyperparameters.

49. ***To Reviewer 4:*** **Q1**: *The use of the property H(cX)=H(X)+log(|c|).* **A1**: We are aware that this does not apply when $c$ is
50. dependent on $X$. The cause of this gap is that we let $m_i$ (i.e. chosen factors) be dependent on $s$, while in section 3 $s_i$ is
51. fixed. If we dig deeper, the root of this gap is that features can not be viewed as factors. A factor could be x coordinate
52. of the agent, but without additional supervision it is impossible for networks to extract such compact information. One
53. way to view features is to see them as index-varying factors. E.g., at timestep $t$ a feature could be $\{x_1,x_2,x_3\}$ but at
54. timestep $t+1$ it could be $\{x_3,x_1,x_2\}$. Then we can let $m_i$ be fixed and introduce a permutation matrix $P(s)$ that is
55. dependent on $s$ and let sub-state $s_i = m_i P(s) \odot f(s)$. It is easy to show that $H(m_i P(X) \odot X) = H(X) + log(|c|)$.
56. However, we did not implement the permutation form in our paper, mainly due to that there are still flaws in the
57. index-varying factor perspective of features and that current RD$^2$ has already achieved significant performance.

[Meta-Review · NeurIPS 2020]

The authors propose an automatic reward decomposition method that allows better credit assignment. The reviewers agree that the approach is interesting and intuitive, and experimental results are positive and include interesting games. The rebuttal was very helpful in clarifying questions raised. Please make sure that you include these clarifications as well as the extra results in the final version of the paper.